# Patient-Guided Talking Points to Address COVID-19 and General Vaccine Hesitancy

**DOI:** 10.3390/pharmacy10050137

**Published:** 2022-10-20

**Authors:** Elaine Nguyen, Melanie Wright, Cathy Oliphant, Kevin Cleveland, John Holmes, Mary Nies, Renee Robinson

**Affiliations:** 1Department of Pharmacy Practice and Administration, College of Pharmacy, Idaho State University, Meridian, ID 83642, USA; 2Department of Pharmacy Practice and Administration, College of Pharmacy, Idaho State University, Pocatello, ID 83209, USA; 3College of Health, School of Nursing, Idaho State University, Pocatello, ID 83209, USA; 4Department of Pharmacy Practice and Administration, College of Pharmacy, Idaho State University, Anchorage, AK 99508, USA

**Keywords:** vaccination, vaccine coverage, vaccination refusal, anti-vaccination movement, COVID-19 vaccines

## Abstract

Vaccination remains one of the most effective ways to limit spread of disease. Waning public confidence in COVID-19 vaccines has resulted in reduced vaccination rates. In fact, despite vaccine availability, many individuals choose to delay COVID-19 vaccination resulting in suboptimal herd immunity and increased viral mutations. A number of qualitative and quantitative studies have been conducted to identify, understand, and address modifiable barriers and factors contributing to COVID-19 vaccine hesitancy among individuals with access to vaccine. Vaccine confidence may be improved through targeted patient–provider discussion. More patients are turning to pharmacists to receive their vaccinations across the lifespan. The primary goal of this commentary is to share evidence-based, patient talking points, tailored by practicing pharmacists, to better communicate and address factors contributing to vaccine hesitancy and reduced vaccine confidence.

## 1. Vaccine Hesitancy

Vaccination remains one of the most effective ways to limit spread of disease, reducing both mortality and morbidity. Waning public confidence in safety and efficacy of vaccinations, especially COVID-19 vaccines, has been shown to foster complacency and hesitancy among patients, reducing immunization uptake and vaccination rates [1,2,3]. According to Health Care Cost Institute (HCCI) claims data, during the COVID-19 pandemic, routine immunizations declined by 18%, posing a major public health threat [4]. Despite vaccine availability, many individuals choose to delay or not receive a COVID-19 vaccine until safety and efficacy are proven against emerging virus strains (e.g., alpha, beta, delta, and omicron variants), resulting in suboptimal herd immunity and increased viral mutations [1,2,3,5].

A number of qualitative and quantitative studies have been conducted to identify, understand, and address modifiable barriers and factors contributing to COVID-19 vaccine hesitancy among individuals who have access to vaccine(s) [6,7,8,9,10,11,12,13]. Vaccine hesitancy, the delay in acceptance or refusal to receive an immunization(s), is influenced by complacency, convenience, and confidence. Research has shown that patient decisions, such as to receive or not to receive a COVID-19 vaccine, are highly influenced by a number of social and cultural factors (e.g., political ideology, past experiences with health services, family histories, and the moral dilemma between individual autonomy and the greater public health), including how the information is delivered and by whom the information is provided (e.g., healthcare provider, caregiver, family member) [8,10,14,15].

## 2. Medical Decision Making

Medical decision making, the ability of a patient to understand the benefits, risks, and treatment/intervention options, is necessary to make an informed medical care decision [16]. Americans generally have a high level of trust in healthcare providers, greater than that for public authorities and/or the government [17]. Therefore, vaccine confidence may be heightened through targeted patient–provider discussion, directly addressing patient safety and efficacy concerns.

## 3. Role of Pharmacists as Vaccine Providers

More and more patients are turning to pharmacists to receive immunizations across the lifespan [18,19,20,21]. New recommendations from the Advisory Committee on Immunization Practices (ACIP) highlight the abilities of pharmacists to assess, influence, expand, and support patient and caregiver medical decisions [22,23]. It is important for healthcare providers to understand patients’ feelings and experiences and to adequately assess patients’ needs to tailor care. The skill set required for healthcare professionals to provide this “patient-centered care” requires not only scientific knowledge and technical aptitude but also affective qualities or virtues such as compassion and empathy.

Confidence and/or trust, both which are critical for vaccine uptake by patients, are impacted directly by patient–provider communication, and the patient–provider relationship. Patient satisfaction, adherence, engagement, and health outcomes have all been shown to correlate with patient–provider communication and perceived provider empathy. Pharmacists, who are among the most accessible, trusted, healthcare providers, are well-positioned to listen to and address patient concerns related to immunization safety and effectiveness, especially in under-resourced communities. As of 15 September 2022, pharmacists in the Federal Retail Pharmacy Program, a collaboration between the Federal government, states, territories, and 21 national pharmacy partners and independent pharmacy networks nationwide, were responsible for administering more than 266.5 million doses of the COVID-19 vaccine; however, if communication were improved, knowledge better demonstrated, culture taken into consideration, and empathy displayed, the number of all vaccinations administered could be far greater [24,25,26,27,28,29,30].

The primary goal of this paper is to share evidence-based strategies and patient talking points, tailored by practicing pharmacists (Table 1), to better address factors contributing to vaccine hesitancy, delays in vaccination, and reduced vaccine confidence through improved information sharing and communication (JMIR PH in Press). This proposed work builds upon the significant vaccine confidence research and practical application of lessons learned by this multidisciplinary investigative team through research, clinical practice, and training of future healthcare providers.

## 4. Discussion

Over the past year, our research team worked with vaccine-hesitant patients in both Alaska and Idaho to identify and understand the many factors contributing to their hesitancy to receive the COVID-19 vaccine. Grounded by the current scientific literature, published guidelines and concerns voiced by patients, providers, healthcare administrators, and public health leaders in the lay literature, our team co-developed a moderator’s guide which was used to better understand the underlying factors contributing to vaccine hesitancy [2,6,7,8,9,11,27,57]. Seven healthcare provider interviews and six focus group discussions were conducted to determine the factors contributing the most to vaccine hesitancy, including, but not limited to, confidence in the vaccine, immunization convenience, factors contributing to complacency and perceived need to receive the vaccine (Publication Pending). Results from the qualitative work were used to develop a Qualtrics survey, which was completed by 736 patients across Alaska and Idaho (Publication Pending). Concerns identified from our previous work were then used to prioritize and categorize patient information needs (Patient question(s) that focus on…), and needs were subsequently translated to confirm patient understanding (Suggested clarifying statements…) using techniques employed in motivational interviewing (Table 1). For each translated concern a number of evidence-based references were identified and patient-friendly statements generated to address patient fears, safety concerns, and risks (Table 1).

Fear, driven primarily by unaddressed information needs and sharing of misinformation (lay press, social media), remains the primary factor contributing to identified vaccine confidence concerns [5,58,59]. Pharmacists can leverage existing patient relationships to address misinformation, alleviate patient safety concerns, and advocate for improved public health. Our previous work has shown that pharmacists need to clearly and concisely address fears related to both potential and perceived short- and long-term side effects associated with COVID-19 vaccines [11,21]. A framework for communication and standardized language should be used to clarify concerns and fears, such as “It sounds like you are concerned” or “It sounds like you are worried”, to simplify evidence-based responses and communicate risk [60,61].

## 5. Conclusions

To better address factors contributing to vaccine hesitancy, delays in vaccination, and reduced vaccine confidence, pharmacists and pharmacy technicians must understand, address, and effectively communicate evidence-based information to patients to alleviate patient fears. Evidence-based, community, and practice-guided patient information resources can be used by pharmacists to improve patient communication, address vaccine hesitancy concerns, and increase vaccination uptake, especially among under-served and poorly resourced communities where vaccine access remains a concern.

## Figures and Tables

**Table 1 pharmacy-10-00137-t001:** Tailored evidence-based, patient talking points.

Patient Question(s) That Focus on…	Suggested Clarifying Statement and Facts to Address Underling Patient Concern Potential Word Substitutions Provided in Parentheses
How do we know these vaccines are safe?	It sounds like you have some concerns (fears) about receiving the COVID-19 vaccine, can you tell me more about your concerns (fears)? - Safety data milestones are required by the FDA for ALL vaccines [31]; - COVID-19 vaccines go through the same safety tests as other vaccines approved and used in the US, just like other vaccines you have received (e.g., mumps, measles) [31]; - FDA-approved COVID-19 vaccines (e.g., Moderna, Pfizer) have passed ALL required “safety milestones” [25,31]; - As of 12 October 2022, there were over 8502 COVID-19 vaccine and drug studies conducted worldwide to ensure that the vaccine is safe and effective in ALL people [25,31,32]. They include people of all races, ethnicities, genders, and with a number of other health conditions (e.g., diabetes) [25,31,32,33,34,35,36].
If the vaccine is so safe why is it causing side effects?	It sounds like you have some concerns about receiving the vaccine due to potential (short term or long-term) side effects, can you tell me more about your concerns regarding (short term or long-term]) safety? - Vaccines teach our immune system (body’s defense system) how to recognize and fight the virus that causes COVID-19. Sometimes this process can cause symptoms, such as fever. These symptoms are normal and are a sign that the body is building protection. Side effects for ALL FDA approved vaccines are monitored very carefully [31]; - The FDA and CDC will continue to monitor safety using two separate registries (Vaccine Adverse Event Reporting System (VAERS), and Vaccine Safety Datalink (VSD)) and a personalized, confidential, health check, v-safe [37,38,39]; - More than 183. 94 COVID-19 vaccine doses have been safely administered per 100 people in the US between 13 December 2020 and 6 September 2022 [25]. Reports of adverse events to VAERS following vaccination, including deaths, do not necessarily mean that a vaccine caused a health problem [38]. It just means that somebody experienced an event after vaccination. Clinicians continue to review reports of these deaths. COVID-19 vaccines are safe and effective and severe reactions after vaccination are rare. Only ~0.3% of recipients reported serious health problems after vaccination, which are very rare [40]; - Side effects from a vaccine are a good indicator that the vaccine is creating the desired immunity (protection) [30]. About 15% of people developed side-effects (e.g., pain at site of shot, fever, chills, tiredness, and headache) lasting a few days. Anaphylaxis, a severe allergic reaction to any vaccine, is rare. It has occurred at a rate of ~5 cases per one million vaccine doses given [37,38,39]. Healthcare providers can effectively and immediately treat the reaction.
How were the COVID-19 vaccines developed so quickly?	It sounds like you are concerned that COVID-19 vaccines were developed so quickly and that they may not be safe or effective. - Over the past 20 years, scientists have been working on a number of different vaccine technologies (e.g., mRNA vaccines) to prevent viral illness without having to give the person weakened, dead, or live bacteria or virus. This type of vaccine works with the body’s natural defenses to safely teach the body how to develop antibodies to fight the virus. Antibodies help to protect you against COVID-19. COVID-19 vaccines use mRNA technology, which is not new [41]; - All FDA required steps to ensure vaccine safety and efficacy were taken in COVID-19 vaccine development [31]; - However, unlike other vaccines, COVID-19 vaccines were approved more quickly because the government removed much of the red tape, provided money that supported many steps to be done at the same time, and decreased the financial risk to drug companies so they could work on a solution [42].
Will the COVID-19 vaccine cause you to get the virus?	It sounds like you are worried (concerned) that you or your loved ones might get COVID-19 if you receive the COVID-19 vaccine. - None of the COVID-19 vaccines currently in development in the US use the live virus that causes COVID-19. Pfizer and Moderna are mRNA vaccines. mRNA vaccines teach your body to make inactive proteins in order to trigger an immune response, they do not contain a live virus; therefore, they cannot give you COVID [43].
Will receiving the mRNA vaccine alter my DNA?	It sounds like you are worried (concerned) that the mRNA vaccine might alter or change your DNA, impacting your health. - mRNA does not affect or interact with your DNA in any way [44,45].
How do we know that these vaccines are safe during pregnancy (or during nursing)?	It sounds like you are worried (concerned) that it might not be safe for you and/or your baby to receive the COVID-19 vaccine while you are pregnant (nursing, or considering getting pregnant). - Pregnant women have a higher risk of severe illness from COVID than non-pregnant women. If you get the vaccine while you are pregnant it can prevent severe illness or death of the mother and pregnancy complications, such as stillbirth or pre-term delivery [46]; - The safety of COVID-19 vaccine use during pregnancy is closely monitored through two established systems: VAERS and v-safe pregnancy registry [37,38,39]; - Approved COVID-19 vaccines have not been shown to cause health problems for either the mom and/or the baby [47,48,49,50].
Why should I have my kids vaccinated when they don’t typically get the severe illness?	It sounds like you are worried that it might be “riskier” for your kids to get the COVID-19 vaccine than to get COVID-19. - Data for kids continue to be evolving, but we do know that kids can still pass on the disease to others and get sick themselves [51]; - Though kids are often less “sick” when they get COVID, they can still get very ill, requiring hospitalization [52]; - Kids can also develop multisystem inflammatory syndrome after infection with COVID-19 [53].
If I’m at low risk for severe COVID disease, why should I get vaccinated?	It sounds like you think it might be “riskier” to get the vaccine since you are at relatively “lower-risk “of getting COVID-19. - Low-risk doesn’t mean no risk, there is still a risk of developing severe illness [54]; - Even though you may be low risk, those you are around may not be, and being vaccinated can help those we care about to stay safe and healthy [54]; - Although low-risk people may not suffer from the disease as severely as others, they can still transmit the disease to those at higher risk [54].
If I have liver and/or kidney problems, should I get vaccinated?	It sounds like you are worried about the potential impact of the vaccine on your kidneys and/or liver. - People with kidney and or liver problems are at a higher risk of getting very sick from COVID-19 [55,56]; - Vaccines are recommended to people with kidney and/or liver disease by National Kidney Foundation and American Liver Foundation [55,56]; - Getting the vaccine will decrease the risk of serious complications related to COVID-19, hospitalization, and death [55,56].

## Data Availability

Not applicable.

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
