# Peer review of "Patient-Guided Talking Points to Address COVID-19 and General Vaccine Hesitancy"

_pharmacy, 2022, doi:10.3390/pharmacy10050137_

Round 1

Reviewer 1 Report

I found the title and the abstract of this paper intriguing, but was disappointed by the content.   I expected more information on how the information presented in the table was selected, and more on what it was presented as it was.  I found the set up of the table distracting - the answer headings didn't really match the questions (they did in focus but they were not worded as an answer) and the talking points weren't consistently worded in a way that patients would be able to understand.  I had expected that this paper might present some "tried and true" wording that practicing pharmacists had developed and found to be effective, but the table fell short of that.  If one were to read it aloud, this weakness becomes even more apparent.  It would also be useful to have some overarching guidelines (e.g., "check that you understand the concern" "avoid jargon such as VAERS").   Some discussion to close the paper is also warranted.   Finally, I would recommend updating the talking points, as I expect that now pharmacists are facing questions about boosters, and variants.  

Author Response

Thank you again reviewers for your thoughtful suggestions. Below you will find the point by point responses to each comment.

General: A brief discussion was added as requested

Reviewer 1: 

Comment 1: I expected more information on how the information presented in the table was selected, and more on what it was presented as it was. I found the set up of the table distracting - the answer headings didn't really match the questions (they did in focus but they were not worded as an answer) and the talking points weren't consistently worded in a way that patients would be able to understand. Response: The talking points were based on questions pharmacists who participated in vaccine hesitancy focus groups (Separate manuscript) and Advisory Board suggestion and review. To decrease and condense the information we grouped questions by category, added a statement to clarify the patient's question, and provided specific facts that could be shared with patients related to the question topic. To make this clearer we have changed table headings to: “Patient questions that focus on…; Suggested Clarifying Statements; and Facts to Address Underlying Patient Concerns”. 

Comment 2: I had expected that this paper might present some "tried and true" wording that practicing pharmacists had developed and found to be effective, but the table fell short of that. Response:  We intended the “Suggested Clarifying Statements” to serve this role and have reviewed each statement, adding additional suggested words in parentheses that may be substituted.

Comment 3: It would also be useful to have some overarching guidelines (e.g., "check that you understand the concern" "avoid jargon such as VAERS").  Response: Registries and tools have been referenced and additional guideline references added throughout paper (See Track Changes) 

Comment 4: Some discussion to close the paper is also warranted. Response: A discussion section has been added as suggested.

Comment 5: Finally, I would recommend updating the talking points, as I expect that now pharmacists are facing questions about boosters, and variants. Response: Talking points have been upd

Reviewer 2 Report

The title of the article should refer specifically to COVID-19 vaccination hesitancy, as this is what the authors consider.

Lines 14-15 refer to vaccination writ large. However, the topic under examination is COVID-19 vaccination. The authors should refer specifically to COVID-19 vaccination.

Lines 15-16: has confidence in COVID-19 vaccines waned? I am unaware of this. Please cite two reputable references. (I assume there are many.)

Lines 32-34: I'm not sure the point of this sentence. Confidence in COVID-19 vaccines has diminished, which has lead to greater COVID-19 vaccine hesitancy, which has lead to reduced COVID-19 vaccine uptake? If this is the point, it seems to be obvious. Reduced confidence in a treatment will lead to lesser uptake. Increased hesitancy for a treatment will lead to reduced uptake. Further, the three references [1-3] do not discuss COVID-19 vaccination.If the authors wish to address vaccination in general (for diseases beyond COVID-19), they should make this more clear; however, if they choose to do so, they need to carefully relate how vaccine hesitancy more broadly applies to vaccine hesitance specific to COVID-19. Clearly vaccine hesitancy differs among various diseases, types of vaccines, points in time, etc.

Lines 34-36 cite [2-6] to say that routine immunizations declined by 18% in 2020. I was unable to find this fact in any of the five references. One reference would suffice to verify the claim; five is rather dubious.

Line 36-39: I cannot find in these references any mention that people are hesitant because they are worried that vaccines will not protect against new variants.

Lines 42-45: I recommend the authors also see https://doi.org/10.1016/j.pmedr.2022.101903 which seems relevant. This paper also contains other valuable references, specifically in Section 4 titled "Discussion".

Line 47-48: I cannot find this result in reference [16]. Please review all your references and ensure they are correct.

Lines 48-50: Again, the references do not seem related to the statement.

Lines 57-58: what are these references? Are they authored by any of the present authors?

Author Response

Thank you again reviewers for your thoughtful suggestions. Below you will find the point by point responses to each comment.

General: A brief discussion was added as requested

Reviewer 2: 

Comment 1: The title of the article should refer specifically to COVID-19 vaccination hesitancy, as this is what the authors consider. Response: We have changed the title as suggested to “Patient-guided Talking Points to Address COVID-19 and General Vaccine Hesitancy”.

Comment 2: Lines 14-15 refer to vaccination writ large. However, the topic under examination is COVID-19 vaccination. The authors should refer specifically to COVID-19 vaccination. Response: We have added the clarification as requested.

Comment 3: Lines 15-16: has confidence in COVID-19 vaccines waned? I am unaware of this. Please cite two reputable references. (I assume there are many.)  Response: We have added references as requested and added clarification to line 30-32 as requested.

Comment 4: Lines 32-34: I'm not sure the point of this sentence. Confidence in COVID-19 vaccines has diminished, which has led to greater COVID-19 vaccine hesitancy, which has led to reduced COVID-19 vaccine uptake? If this is the point, it seems to be obvious. Reduced confidence in a treatment will lead to lesser uptake. Increased hesitancy for a treatment will lead to reduced uptake. Further, the three references [1-3] do not discuss COVID-19 vaccination.If the authors wish to address vaccination in general (for diseases beyond COVID-19), they should make this more clear; however, if they choose to do so, they need to carefully relate how vaccine hesitancy more broadly applies to vaccine hesitancy specific to COVID-19. Clearly vaccine hesitancy differs among various diseases, types of vaccines, points in time, etc. Response: We do appreciate that these connections between confidence, hesitancy, and vaccine uptake are clear to many but potentially not all readers. We added safety and efficacy to the previous sentence to connect the reason but think that inclusion of this review statement is important to frame the message. 

Comment 5: Lines 34-36 cite [2-6] to say that routine immunizations declined by 18% in 2020. I was unable to find this fact in any of the five references. One reference would suffice to verify the claim; five is rather dubious. Response: We have removed all but one reference as requested - The Impact of COVID-19 on the Use of Preventive Health Care - HCCI. Accessed March 1, 2022. https://healthcostinstitute.org/hcci-research/the-impact-of-covid-19-on-the-use-of-preventive-health-care

Comment 6: Line 36-39: I cannot find in these references any mention that people are hesitant because they are worried that vaccines will not protect against new variants.  Response: Thank you for the clarification request. We have added an additional reference that explicitly states the connection rather than inferring connection. We apologize for the zotero referencing issues. Mallapaty S. Researchers fear growing COVID vaccine hesitancy in developing nations. Nature. 2022 Jan;601(7892):174-175. doi: 10.1038/d41586-021-03830-7. PMID: 34949861.

Comment 7: Lines 42-45: I recommend the authors also see https://doi.org/10.1016/j.pmedr.2022.101903 which seems relevant. This paper also contains other valuable references, specifically in Section 4 titled "Discussion". Response: I think there may have been a copy and paste issue. The suggested reference entitled “Multisociety Consensus Quality Improvement Revised Consensus Statement for Endovascular Therapy of Acute Ischemic Stroke” does not appear to be relevant and was not added.

Comment 8: Line 47-48: I cannot find this result in reference [16]. Please review all your references and ensure they are correct. Response: We have reviewed and adjusted references, apologize for confusion that resulted when putting from zotero reference manager program into the template.

Comment 9: Lines 48-50: Again, the references do not seem related to the statement. Response: We have reviewed and adjusted references, apologize for confusion with reference manager program.

Comment 10: Lines 57-58: what are these references? Are they authored by any of the present authors? Response: The included references highlight the role of pharmacists as immunizers, present authors were not involved in any of these publications.